# Endothelial Cell Plasma Membrane Biomechanics Mediates Effects of Pro-Inflammatory Factors on Endothelial Mechanosensors: Vicious Circle Formation in Atherogenic Inflammation

**DOI:** 10.3390/membranes12020205

**Published:** 2022-02-10

**Authors:** Nadezhda Barvitenko, Mohammad Ashrafuzzaman, Alfons Lawen, Elisaveta Skverchinskaya, Carlota Saldanha, Alessia Manca, Giuseppe Uras, Muhammad Aslam, Antonella Pantaleo

**Affiliations:** 1Independent Researcher, 191014 Saint-Petersburg, Russia; 2Department of Biochemistry, College of Science, King Saud University, Riyadh 11451, Saudi Arabia; mashrafuzzaman@ksu.edu.sa; 3Department of Biochemistry and Molecular Biology, School of Biomedical Sciences, Monash University, Melbourne, VIC 3800, Australia; alfons.lawen@monash.edu; 4Sechenov Institute of Evolutionary Physiology and Biochemistry, 194223 Saint-Petersburg, Russia; lisarafail@mail.ru; 5Institute of Biochemistry, Institute of Molecular Medicine, Faculty of Medicine, University of Lisbon, 1649-028 Lisboa, Portugal; carlotasaldanha@fm.ul.pt; 6Department of Biomedical Science, University of Sassari, 07100 Sassari, Italy; alessia_manca@hotmail.it; 7Department of Clinical and Movement Neurosciences, Institute of Neurology, University College London, London NW3 2PF, UK; g.uras@ucl.ac.uk; 8Experimental Cardiology, Department of Cardiology/Angiology, Justus Liebig University, 35392 Giessen, Germany; muhammad.aslam@physiomed.jlug.de

**Keywords:** shear stress, inflammation, endothelial cell, lipid bilayer, cytoskeleton, mechanosensor, atherosclerosis, oxidative stress

## Abstract

Chronic low-grade vascular inflammation and endothelial dysfunction significantly contribute to the pathogenesis of cardiovascular diseases. In endothelial cells (ECs), anti-inflammatory or pro-inflammatory signaling can be induced by different patterns of the fluid shear stress (SS) exerted by blood flow on ECs. Laminar blood flow with high magnitude is anti-inflammatory, while disturbed flow and laminar flow with low magnitude is pro-inflammatory. Endothelial mechanosensors are the key upstream signaling proteins in SS-induced pro- and anti-inflammatory responses. Being transmembrane proteins, mechanosensors, not only experience fluid SS but also become regulated by the biomechanical properties of the lipid bilayer and the cytoskeleton. We review the apparent effects of pro-inflammatory factors (hypoxia, oxidative stress, hypercholesterolemia, and cytokines) on the biomechanics of the lipid bilayer and the cytoskeleton. An analysis of the available data suggests that the formation of a vicious circle may occur, in which pro-inflammatory cytokines enhance and attenuate SS-induced pro-inflammatory and anti-inflammatory signaling, respectively.

## 1. Introduction

### 1.1. Blood Flow Patterns and the Pro-Inflammatory Response of Endothelial Cells

Inflammation is a key mechanism affecting endothelial cells (ECs) and leading to atherosclerosis [1,2,3,4]. ECs are subjected to three main hemodynamic forces: hydrostatic pressure, cyclic stretch, and shear stress (SS). Fluid SS is a frictional force, produced by blood flow, acting tangentially to the surface of the ECs. Depending on the patterns of fluid SS acting on ECs, either anti- or pro-inflammatory mechanisms can be triggered in ECs. High laminar SS induces an anti-inflammatory response [2,5,6], while low laminar SS and disturbed patterns of SS, on the other hand, activate pro-inflammatory mechanisms in ECs [2,3]. The key transcription factors responsible for the low flow-induced inflammatory response of ECs are activator protein 1 (AP-1), nuclear factor κB (NF-κB) [3], and yes-associated protein/transcriptional coactivators with a PDZ-binding motif (YAP/TAZ) [7]. The transcription factors Krüppel-like factor 2 (KLF2), myocyte enhancer factor 2 (MEF2) and nuclear factor erythroid 2-related factor 2 (NRF2) are known to activate an anti-inflammatory response [3,6].

The molecular pathways leading to inflammation in ECs begin at the endothelial mechanosensors, one of which is the platelet endothelial cell adhesion molecule-1 (PECAM-1), which forms a mechanosensory complex with vascular endothelial cadherin (VE-cadherin) and vascular endothelial growth factor receptor 2 (VEGFR2). The PECAM-1/VE-cadherin/VEGFR2 complex triggers the activation of the nuclear factor κB (NF-κB) transcription factor [8,9]. In this complex, PECAM-1 transmits the mechanical signal, VE-cadherin is an adaptor protein, and VEGFR2 activates phosphatidylinositol-3-OH kinase (PI3K) [8]. The activation of the PECAM-1/VE-cadherin/VEGFR2 complex by SS (12 dyn/cm^2^) occurs within 15 s, leading to the activation of integrin and integrin-dependent EC alignment in the direction of flow, and the transient activation of NF-κB [8]. It should be noted that the signaling triggered by VEGF via its receptors can exhibit both pro-inflammatory and vasculo-protective anti-inflammatory actions. For example, VEGF participates in angiotensin II (Ang II)-induced vascular inflammation [10], while the local application of recombinant VEGF or the local VEGF gene transfer to arterial wall ECs in the hind-limb ischemia model enhanced vascularization and collateral circulation [11,12]. In mouse aortic ECs (MAECs) and bovine aortic ECs (BAECs), high laminar SS (24 dyn/cm^2^) activates integrins and NF-κB [9]. The seemingly paradoxical nature of that high laminar SS-activated pro-inflammatory NF-κB can be explained via transient NF-κB activation (in the case of high laminar SS) versus sustained NF-κB activation (in the case of oscillatory or disturbed SS) [9]. In human aortic ECs (HAECs), the activation of αvβ3 integrins by oscillatory flow promotes NF-κB activation and atherogenesis [13]. Activated NF-κB induces the expression of pro-inflammatory cytokines, chemokines, and cell adhesion molecules [14,15]. Likewise, atheroprotective laminar SS induces the inactivation of pro-inflammatory YAP/TAZ via their phosphorylation-mediated cytoplasmic retention in ECs [7]. On the other hand, oscillatory SS results in the hyperactivation of YAP/TAZ via RhoA/Rock-dependent actin stress fiber formation, leading to the enhanced expression of inflammatory ICAM1 and VCAM1 [7,16]. The endothelial specific over-expression of YAP exacerbates, while CRISPR-mediated YAP-knockdown retards, plaque formation in ApoE^−/−^ mice [16,17].

These links between blood flow patterns, the mechanosensitivity of ECs, and inflammation evoked deep interest in the study of endothelial mechanosensors [18,19,20,21,22,23,24,25,26,27], which include primary cilia, glycocalyx, integrins, caveolae, ion channels, heterotrimeric G proteins, G protein-coupled receptors (GPCRs), PECAM-1, VE-cadherin, VEGFR2, and the Tie family of receptor tyrosine kinases [20]. Moreover, Notch1 and guidance receptor plexin D1 can also function as endothelial mechanosensors [28,29]. Likewise, in human pulmonary aortic ECs (HPAECs), mitochondria respond to fluid SS with increased oxidative phosphorylation and elevated ATP production [30,31], suggesting their possible role as mechanosensors.

### 1.2. Forces and Plasma Membrane Mechanosensors

Two models have been proposed for the mechanical force-induced activation of mechanosensitive ion channels (Figure 1): “force-from-lipids” and “force-from-filament” [32]. The “force-from-lipid” model suggests the gating of mechanosensitive ion channels with inputs from the lipid bilayer, while the “force-from-filament” model suggests primary roles of the extracellular matrix and intracellular cytoskeleton (CSK) in the mechanical activation of ion channels [32]. Great efforts have been devoted to deciphering the multiple effects exerted by membrane lipids on the structure and function of transmembrane proteins [33,34,35,36,37,38,39,40,41,42,43,44].

Two major types of physical effects, integral membrane proteins or ion channels, are expected to draw from the hosting lipid bilayer due to its profiles of electrical charges and mechanical properties [43]. The former one arises due to the consideration that the distribution of the charges of the lipids on either monolayer may or may not show any net charges, but due to their coupling with the integral membrane proteins (MPs) or channel proteins, the charges on both lipids and proteins become redistributed or polarized. As a result, any MP-lipid coupling appears with a distinctive type of physical phenomenon, many charges interactions, as explained using screened Coulomb interactions (SCIs) in ref. [43]. Considering the mechanical properties (bilayer elasticity and lipid intrinsic curvature) of lipid layers, only the bilayer regulation of the integral membrane protein function has long been addressed (see refs. [45,46,47,48,49,50,51,52]). However, in refs. [43,44], it is clearly shown that the SCI model that considers the charge-based interactions among integral MPs and hosting bilayer lipids can correctly address the bilayer regulation of MP or channel functions, and that the charge-based interactions appear to be primary regulators of channel functions. The bilayer mechanical property-based regulation of MP functions also appears in SCI treatment, but only to produce a secondary effect on MP functions. In the elastic bilayer model, the primary effect due to charge-based interactions was totally ignored [45,46,47,48,49,50,51,52]. The charge-based effects have later been consistently found to be appearing as primary molecular mechanisms, especially when using molecular dynamics (MD) simulations, on varieties of membrane-adsorbed peptides and drugs cases (see refs. [53,54]).

Recently, the plasma membrane of immune cells was suggested to integrate multiple biophysical and biochemical stimuli (such as cholesterol content, negatively charged lipids, electrical potential) in order to regulate immune receptor function [55].

Both the composition of the lipid bilayer [22] and the NMMII-generated basal tension of EC [56] can be altered by pro-inflammatory stimuli. Endothelial mechanosensors, which are embedded in the plasma membrane, not only influence the blood flow, but also regulate the mechanical properties of the lipid bilayer and CSK. Here, we discuss if and how pro-inflammatory factors may change the mechanical properties of the lipid bilayer and CSK. The rigidification of the lipid bilayer and the increase in the cytoskeletal NMMII-generated tension can increase the energy barrier for the activation of endothelial mechanosensors by SS. As a result, high SS can be perceived by ECs as low SS. Since low SS induces the release of pro-inflammatory mediators [2,4] the *vicious circle* can be formed. It keeps up the low-grade vascular inflammation and promotes the development of endothelial dysfunction and atherosclerosis. Recently, the primary cilia on ECs were suggested to amplify low unidirectional SS signaling, resulting in the activation of NRF2 and the protection of ECs from oxidative damage [6]. We discuss the data suggesting that, under inflammatory conditions, the lipid bilayer and the NMMII-generated tension could dampen high laminar SS.

## 2. What Are the Intracellular Forces Acting on Any Single Transmembrane Endothelial Mechanosensor?

### 2.1. From Stiffness of the Whole EC to the Mapping of Intracellular Forces Acting on Single Transmembrane Mechanosensor: From Cell- to Protein-Scale Studies

Blood flow exerts extracellular forces, such as hydrodynamic pressure, cyclic stretch, and fluid SS. In addition to these forces, the substrate stiffness is also sensed by ECs [57,58]. The dependence of the mechanical properties of the cortical CSK in HAECs and human umbilical vein ECs (HUVECs) on laminar SS strength was investigated using acoustic force spectroscopy [59]. The exposure of HAECs and HUVECs to laminar SS (6 dyn/cm^2^ for up to 48 h) was found to evoke an increase in the membrane cortex stiffness [59].

The stiffness of the whole EC, or its plasma membrane with the underlying submembrane actin-based CSK (smACSK), is an integral parameter, evaluating the behavior of the whole cell. At the cellular level, the stiffness of bovine pulmonary arterial ECs (BPAECs) depends on the basal isometric tension, which is determined by NMMII contractility [56]. Thrombin induces a rapid increase in basal isometric tension in BPAECs via the MLCK- and RhoA-mediated activation of NMMII [56]. Further studies on the biomechanics of whole ECs should lead to an analysis of the spectrum of intracellularly generated forces that converge on any single transmembrane mechanosensor [60]. Evidence is accumulating that forces generated within the cell regulate mechanical tension across the transmembrane, cytoskeletal, and scaffolding proteins [61,62,63].

Endothelial mechanosensors are subjected to forces generated within the cell, in particular within the lipid bilayer of the plasma membrane, and forces arising from smACSK (Figure 2). Energy inputs from the lipid bilayer and the smACSK are likely to increase, or decrease, the activation energy required for the stimulation of mechanosensors by fluid SS (Figure 2). Earlier, we proposed that the gradients in hydrostatic pressures across the plasma membrane induced by changes in cell volume are actively probed by cells via the pulling activity of non-muscle myosin II (NMMII) and the pushing activity of smACSK [64]. Considering the forces arising from the lipid bilayer (see Section 1.2), and the forces generated by smACSK, the energy input (E(intracellular)) that is received—in addition to extracellular mechanical forces—by any single mechanosensory can be presented as the sum of the following energies:E(intracellular) = E(lipid bilayer) + E(NMMII) + E(protrusion) + E(resistance) + E(smACSK spring)(1)
where E(lipid bilayer) is an energy from the lipid bilayer, E(NMMII) is an energy of NMMII-generated pulling (directed into the cell) force, E(protrusion) is an energy of pushing (directed out of the cell) force due to the actin-based assembly of lamellipodia and filopodia, E(resistance) is an energy generated by the lipid bilayer together with smACSK, and E(smACSK spring) is an energy stored by smACSK during its mechanical deformation.

Thus, the energies received from within the cell would also influence the activation of an endothelial mechanosensor:E(activation of endothelial mechanosensor) = E(extracellular) + E(intracellular)(2)

In this paper, we only pay attention to three intracellular forces: the biomechanics of the lipid bilayer, the pulling force of NMMII, and the pushing force of the lamellipodia. NMMII and smACSK are controlled by many intracellular signaling mechanisms; however, we only consider two functionally antagonistic signaling proteins, RhoA and Rac1. The convergence of several pro-inflammatory stimuli on the two functionally antagonistic small GTPases RhoA and Rac1 is discussed in Section 3.

### 2.2. Biomechanics of the Lipid Bilayer and the Activation Energy of Mechanosensors

There are two mechanisms for the regulation of transmembrane proteins by lipids: a ligand-like mechanism, when the direct high-affinity binding of lipids to proteins occurs, and a solvent-like mechanism, when the addition or removal of a lipid changes the biomechanics of the lipid bilayer [65,66]. For example, many transmembrane proteins contain specific motifs for cholesterol binding: a cholesterol recognition/interaction amino acid consensus (CRAC, R/K-X_5_-Y-X_5_-L/V), a reversed CRAC motif named CARC (L/V-X_5_-Y-X_5_-R/K), and a cholesterol consensus motif (CCM) [65,66,67,68,69]. In this paper, we mainly pay attention to the solvent-like scenario, which deals with alterations in plasma membrane biomechanics.

There is convincing evidence that the lipid composition, which determines the lipid bilayer’s fluidity, influences both basal and SS-induced GTPase activities of G_αq_ and G_αi3_ subunits of heterotrimeric G proteins in phospholipid vesicles [70]. The incorporation of lysophosphatidylcholine into liposomes increases the fluidity of the lipid bilayer and elevates the basal activity of G_αq_ and G_αi3_ proteins from 0.47 to 1.35 pmol/min per μg of protein [70]. The incorporation of benzyl alcohol, another fluidizing agent, increases the basal activity of G proteins from 0.47 to 2.37 pmol/min per μg of proteins. On the other hand, the incorporation of cholesterol, which decreases bilayer fluidity, diminishes the basal activity of Gαq and Gαi3 proteins from 0.47 to 0.113 pmol/min per μg of protein and reduces the activation of G proteins by SS [70].

In HUVECs, fluid SS (from 0.7 to 33 dyn/cm^2^) induces an increase in membrane fluidity [71]. The addition of benzyl alcohol also increases the membrane fluidity [71]. The exposure of BAECs to fluid SS increases the plasma membrane fluidity [72]. Additionally, BAECs’ membrane fluidity is increased and decreased by benzyl alcohol (a fluidizing agent) and cholesterol (a rigidifying agent), respectively [72]. In HPAECs, fluid SS increases membrane fluidity [30,31]. Furthermore, the plasma membranes in HPAECs discriminate between cyclic stretch and fluid SS, in that cyclic stretch increases the lipid bilayer order and decreases fluidity, while fluid SS decreases the lipid bilayer order and increases fluidity [73].

Due to all the changes in the membrane composition, which have been explained above, a membrane’s two major physical properties, namely, the charge profiles and mechanical properties, may especially become altered. Consequently, as explained earlier, the membrane regulation of integral MP functions also changes [43,45,46,47,48,49,50,51,52]. However, the exact energy (generally refereed as the ‘free energy of bilayer–integral protein coupling’) that plays important roles in such membrane regulation of integral protein functions has been correctly calculated using the SCI models that consider charge-based interactions [43,53,54]. The SCI model explains all the parameters behind calculating the free energy of bilayer–integral protein coupling, and that this energy consists of both components drawn from the charge properties and mechanical properties of the bilayer and MPs. Thus, it appears to be a universal mechanism which also raises some universal probability functions related to any bilayer-MP coupling energetics (see details in ref. [54]). MD simulations on lipid–drug pair interactions in the bilayer environment have especially demonstrated these universal probability functions to be primarily relying on two major types of charge-based lipid–drug interactions, namely the electrostatic and van der Waals interactions.

### 2.3. Mechanosensors and Force Generated by NMMII

#### 2.3.1. Control of NMMII Contractility

NMMII is an actin-based heterohexameric molecular motor consisting of two heavy chains (HCs), two essential light chains (ELCs), and two regulatory light chains (RLCs) [74,75,76,77,78]. There are three isoforms of NMMII depending on HC paralog: NMMIIA, NMMIIB, and NMMIIC [74,75,76,77], which apparently have partially distinctive roles [79]. In ECs, mainly NMMIIA and NMMIIB are expressed [80,81].

The phosphorylation of the Ser19/Thr18 of RLC by Ca^2+^/calmodulin-dependent myosin light chain kinase (MLCK) activates NMMII contractility (Figure 3) [82], whereas the dephosphorylation of this site by myosin light chain phosphatase (MLCP) inhibits NMMII [83,84]. MLCK can itself be activated by Ca^2+^/calmodulin, protein tyrosine kinases (PTKs), which phosphorylate Tyr464 and Tyr471, and protein kinase C (PKC), while phosphorylation by protein kinase A (PKA) inhibits MLCK [85,86].

The small GTPases Rho, Rac, and Cdc42, govern the formation of actin stress fibers, lamellipodia, and filopodia, respectively [87]. In humans, there are 20 members in the Rho family which are subdivided into subfamilies: Rho, Rac, Cdc42, RhoU/V, RhoD/F, Rnd, RhoH, and RhoBTB [88]. The Rho subfamily consists of RhoA, RhoB, and RhoC, while the Rac subfamily includes Rac1, Rac2, Rac3, and RhoG [88]. In the vasculature, the small GTPases Rho, Rac, and Cdc42, control a number of functions, including the maintenance of the endothelial barrier, the response to SS, the regulation of endothelial nitric oxide synthase, migration, and apoptosis [89]. There is reciprocal regulation of RhoA and Rac1. For example, in BAECs, the activation of integrins by laminar SS (12 dynes/cm^2^) transiently inhibits Rho, but activates Rac1 [90,91]. The small GTPases, RhoA and Rac1, are particularly well-studied as regulators of the endothelial barrier function, where RhoA and Rac1 activation leads to barrier disruption and stabilization, respectively [85,92,93,94,95].

The small GTPase, RhoA, and its effector, Rho-associated coiled-coil-containing kinase (ROCK), activate NMMII, both via the phosphorylation of RLC at Ser19/Thr18 [96] and the phosphorylation and inhibition of MLCP [97]. ROCK, the main downstream effector of RhoA, has two isoforms: ROCK1 and ROCK2 [98]. Rac1 and its effector, p21-activated kinase (PAK), inhibit NMMII via the phosphorylation of MLCK by PAK [99]. Rac1 itself can be regulated by cell-generated tension: in rat aortic smooth muscle cells, the inhibition of myosin contractility via the inhibition of Rho-kinase with Y-27632 or MLCK with ML-7 increased Rac1 activity [61].

Seemingly, the RhoA-dependent activation of NMMII and the generation of the centripetal force, as well as Rac1-dependent protrusive actin polymerization, would affect the endothelial mechanosensors. As pro-inflammatory agents, via the activation of RhoA and/or Rac1, they not only regulate the endothelial permeability, but also may tune mechanosensors because of the inducing pulling (RhoA-mediated NMMII contractility) or pushing (Rac1-mediated actin-based protrusions) of intracellular forces.

#### 2.3.2. Opposing Actions of RhoA and Rac1 on NMMII-Generated Pulling Force Acting on VE-Cadherin

Transmembrane proteins experience an NMMII-generated force. Föster resonance energy transfer (FRET)-based molecular tension sensors allow measuring the pico-Newton (pN) forces acting on cellular proteins [62,100] (Table 1). In static BAECs, vinculin was shown to be under an NMMII-generated tensile force of about 2.5 pN (0.25 μdyn) [62]. In Madin-Darby canine kidney (MDCK) epithelial cells, epithelial cadherin (E-cadherin) was under a constitutive 1–2 pN (0.1–0.2 μdyn) of tensile force generated by NMMII [100]. In static BAECs and those experiencing SS, VE-cadherin was under a tension of 2.4 and 1.8 nN (0.24 and 0.18 mdyn)/molecule, respectively [63]. The tension across PECAM-1 in static BAECs was negligible, but increased under SS tension to 2.0 pN (0.2 μdyn)/molecule in a vimentin-dependent manner [63].

Generally, RhoA and Rac1 increase and attenuate the NMMII contractile force acting on VE-cadherin, respectively (Figure 4). For example, in human dermal microvascular ECs (HMECs) and human pulmonary arterial ECs (HPAECs), the counterbalance between Rho and Rac1 determines the force pulling the VE-cadherin into the cell interior [101]. RhoA activation increases the NMMII-generated tension on VE-cadherin, while Rac1 decreases it [101].

#### 2.3.3. Rac1 in the Regulation of Actin Polymerization Pushing Force in Lamellipodia

ECs are known to form lamellipodia, which are protrusive actin-based structures [102]. Lamellipodia assembly is mainly governed by Rac1 [87,88]. Lamellipodia push the plasma membrane out from the cell interior, and this pressure can increase or decrease the energy required for the activation of any endothelial mechanosensory by SS. For example, VE-cadherin, an element of the mechanosensory PECAM-1/VE-cadherin/VEGFR2 complex [8], is a component in lamellipodia in HUVECs [102]. There is an interesting interplay between pushing, which is generated by protrusive actin polymerization, and NMMII-dependent pulling forces in the formation of VE-cadherin mediated adherens junctions in HUVECs [102]. It should be noted that NMMII contractility is required for lamellipodia formation [102,103].

## 3. Effects of Pro-Inflammatory Stimuli on the Biomechanics of the Lipid Bilayer and Submembrane Cytoskeleton; Focus on Counterbalance between RhoA and Rac1

### 3.1. Pro-Inflammatory Stimuli and the Lipid Bilayer Biomechanics

Hypoxia itself can be a mechanical signal for ECs, as a decrease in the number of dioxygen molecules dissolved in the lipid bilayer of the plasma membrane is a mechanical stimulus that can influence the mechanosensitive transmembrane proteins and, thus, participate in the hypoxia response [104].

However, hypoxia can also lead to the increased production of reactive oxygen species (ROS) [105], which may increase membrane lipid peroxidation, with effects on the mechanical properties of the lipid bilayer. Oxysterols, products of cholesterol oxidation, promote the development of atherosclerosis [106]. Connections between dyslipidemia and the mechanical properties of ECs were reviewed elsewhere [22]. Hypercholesterolemia leads to the accumulation of cholesterol in ECs, and promotes the development of inflammation and atherosclerosis [22,107,108]. An increase in low-density lipoprotein (LDL) cholesterol levels and a decrease in high-density lipoprotein (HDL) cholesterol levels in blood plasma are among the key risk factors for atherogenesis [107,109].

Oxidative stress leads to an accumulation of oxidized phospholipids in the EC plasma membrane [110]. The peroxidation of membrane lipids decreases the lipid bilayer thickness [111,112]. On the other hand, long-chain polyunsaturated fatty acids (PUFAs)—such as eicosapentaenoic acid (EPA, C20:5, n-3), docosahexaenoic acid (DHA, C22:6, n-3), and docosapentaenoic acid (DPA, C22:5, n-3)—trigger anti-inflammatory anti-atherogenic responses in ECs, as can be exemplified with docosahexaenoic acid (DHA) (22:6ω-3) [113,114,115].

### 3.2. Pro-Inflammatory Stimuli in RhoA and Rac1 Regulation in ECs

#### 3.2.1. Hypoxia and Oxidative Stress in the Regulation of RhoA and Rac1 in ECs

In cultured porcine aortic ECs (PAECs), hypoxia induces the activation of RhoA and the inhibition of Rac1 [93] (Table 2). In piglet PAECs, hypoxia activates RhoA and inhibits Rac1 [116]. In rat PAECs, hypoxia activates RhoA [117]. In rat PAECs, oxidative stress (H_2_O_2_) activates RhoA [117]. In BAECs, H_2_O_2_ induces the activation of Rac1 [118].

#### 3.2.2. Pro-Inflammatory Cytokines in RhoA and Rac1 Regulation in ECs

As discussed above, RhoA-ROCK activation via the induction of actomyosin contractility may increase the tension experienced by EC mechanosensors, leading to an increase in the pro-inflammatory response in ECs. Pro-inflammatory mediators unbalance RhoA–Rac1 activities and homeostasis, resulting in changes to the intracellularly generated tension and mechanosensors’ activation threshold.

Ang II can induce vascular inflammation and remodeling [10]. Ang II acts via two types of Ang receptors, type 1 (AT_1_R) and type 2 receptor (AT_2_R), which significantly differ in their physiological effects [119]. Signaling through AT_1_R leads to vasoconstriction, oxidative stress, and inflammation, while signaling through AT_2_R mediates anti-inflammatory effects [120,121] and prevents the development of hypertension in animal models of hypertension [119]. In BAECs, the stimulation of AT_1_R leads to the sequential activation of G_α12/13_ and RhoA [122]. In BAECs, Ang II acting via AT_1_R activates Rac1, and elevates focal adhesion complexes and actin fiber formation [118]. In contrast, the activation of AT_2_R has been linked with the negative regulation of RhoA activity in vascular smooth muscle cells [123]. In BAECs, C-reactive protein (CRP) activates the RhoA–ROCK pathway to induce the expression of plasminogen activator inhibitor-1 [124].

Sphingosine 1-phosphate (S1P) may exert both anti-inflammatory anti-atherogenic effects, when acting through S1P receptor type 1 (S1P_1_ receptor), and pro-inflammatory pro-atherogenic effects, when acting through S1P_2_ and S1P_3_ receptors [125,126,127]. The stimulation of S1P_1_ receptor signaling activates G_αi_ and Rac1, leading to the suppression of the pro-inflammatory response [126]. Both S1P_2_ and S1P_3_ receptors are coupled to G_αi/o_, G_αq_, and G_α12/13_ and their stimulation activates the RhoA–ROCK axis and destabilizes endothelial barrier [125,126,127].

Thrombin, an important regulator of acute and chronic vascular inflammation, acts via protease-activated receptors (PARs), of which there are four isoforms (PAR-1, -2, -3, and -4) [128]. In HUVECs, thrombin activates RhoA, suppresses Rac1 activity, and induces actomyosin contractility [129,130]. Intermedin, a member of the calcitonin gene-related peptide family, and acting via calcitonin receptor-like receptors, antagonizes thrombin-induced endothelial hyperpermeability via the activation of Rac1 [129].

## 4. Some Mechanosensors Are Located in the Plasma Membrane of the EC; Their Sensitivity to the Lipid Bilayer and the CSK Biomechanics

### 4.1. Piezo1

Piezo1 is a transmembrane cation channel [131,132,133] that is gated by membrane tension and SS [134,135]. Full-length *Piezo1* was cloned from the mouse neuroblastoma N2A cell line and expressed in several other cell lines [136]. Piezo1 exhibits activation by stretching [136,137]. Fluorescent Piezo1 constructs were expressed in HEK293 cells [134]. Blebs formed in these transfected cells were deficient in the cytoskeletal proteins, and the basal Piezo1 activity in the bleb-attached patches was higher than in whole-cell-attached patches, suggesting that the membrane tension is a main driver of mechanosensitive gating of Piezo1, whereas the CSK has a mechanoprotective role [134].

Piezo1 performs multiple roles in the cardiovascular system, and links mechanical stimuli to the triggering of both pro- and anti-atherogenic responses in ECs [138]. Piezo1 is expressed in many cell types participating in the development of atherosclerosis, including ECs, vascular smooth muscles cells, T and B cells, and monocytes, which undergo sequential transition into macrophages and lipid-engorged foam cells [138]. The endothelial-specific deletion of Piezo1 in mice impairs SS-mediated vascular development [131], sprouting angiogenesis, and vascular lumen formation [139]. Moreover, EC Piezo1 can sense disturbed blood flow and is linked to inflammatory signaling [140]. In human umbilical arterial ECs (HUAECs), the fluid SS-induced activation of Piezo1 leads to ATP release and autocrine stimulation of the purinergic P2Y_2_ receptor and its downstream effectors, G_αq_ and G_α11_, which triggers the activation of eNOS, NO release, and vasodilatation [141]. Depending on blood flow patterns, Piezo1 activation, together with the activation of purinergic P2Y_2_ receptor and G_αq/11_, may lead to atheroprotective signaling or atherogenic signaling in response to laminar SS or disturbed SS, respectively [140]. In the case of disturbed flow, the induction of atherogenic signaling proceeds via the activation of integrin by SS sensors Piezo1 and P2Y_2_–G_αq/11_ [140].

### 4.2. Mechanosensory PECAM-1/VE-Cadherin/VEGFR2 Complex

VE-cadherin is both a key player in the regulation of the endothelial barrier function [94,95] and an element in the PECAM-1/VE-cadherin/VEGFR2 mechanosensory complex [8]. Interestingly, a direct association between PECAM-1 and G_αq/11_ was reported in HUVECs [142,143]. In HUVECs, the dissociation of G_αq/11_ from PECAM-1 is dependent on G_αq/11_ activation by fluid SS, and on the patterns of fluid SS. Here, impulse and oscillatory, but not ramped-transient flow induced G_αq/11_ dissociation from PECAM-1 [142]. VEGFR2 is localized in lipid rafts, and interference in the lipid raft structure may impair its activation by SS [144]. The transmembrane domains of VE-cadherin and VEGFR2 and VEGFR3 can interact with each other, and this interaction is important for SS signal transduction onto and the activation of VEGFR2/3 [145]. The linking of VE-cadherin to the actin CSK seems to be mediated by β- and α-catenins, as it takes place in epithelial cells containing epithelial (E)-cadherin [146]. E-cadherin binds to β-catenin, β-catenin interacts with α-catenin, and α-catenin binds to actin filaments [146].

The SS response of ECs includes re-arrangement in the EC cytoskeleton and the redistribution of the intracellular forces acting on PECAM-1 and VE-cadherin [63]. The onset of the flow elicited an increase in the tension on PECAM-1 and a decrease in the tension on VE-cadherin [63]. Vimentin, an intermediate filament, appears to transmit NMMII tension onto PECAM-1 [63]. In our hypothesis, RhoA- and Rac1-dependent intracellular forces can tune the mechanosensitivity of the PECAM-1/VE-cadherin/VEGFR2 complex (Figure 5). Indeed, in HUVECs, the VE-cadherin located in the lamellipodia undergoes exhibits both pushing and pulling forces [102]. S1P, via the S1P_1_ receptor, activates Rac1 and suppresses pro-inflammatory elevation in endothelial permeability [126,127] (Figure 5). Rac1 inhibits the RhoA–ROCK pathway that reduces NMMII-generated tension on VE-cadherin, resulting in the stabilization of VE-cadherin [101]. On the other hand, NMMII contractility also contributes to lamellipodia formation [102,103].

### 4.3. Heterotrimeric G Proteins and GPCRs

There is experimental evidence for the activation of GPCRs and heterotrimeric G proteins by fluid SS in the absence of GPCR agonists. The exposure of HUVECs to fluid SS (10 dyne/cm^2^) induces the rapid (within 1 s) activation of G_αq_ and G_αi3_ [147]. The reconstitution of purified G_αq_ and G_αi3_ in phospholipid liposomes reveals that the lipid composition significantly influences the activation of G proteins in response to fluid SS (0–30 dynes/cm^2^) [70]. In human coronary artery ECs (HCAECs), G_αq/11_ proteins are activated by fluid SS independently of upstream GPCRs [148]. In addition, in HCAECs, the stimulation of G_αq/11_ proteins by fluid SS is independent of the Piezo1 channel [143].

The human bradykinin type 2 receptor (B_2_R), with an inserted yellow fluorescent protein and fused to a cyan fluorescent protein, was expressed in BAECs [149]. The recombinant B_2_R was activated by fluid SS, hypotonic stress, and benzyl alcohol in the absence of a B_2_R agonist [149]. Several long-chain polyunsaturated fatty acids (eicosapentaenoic acid, docosahexaenoic acid, docosadienoic acid, and dihomo-γ-linoleic acid) activated B_2_R in BAECs and in B_2_R-transfected HEK293 cells in a ligand-independent manner [150]. The search for a structural motif(s) responsible for the mechanosensitivity of the human histamine type 1 (H_1_) receptor in HUVECs revealed that helix 8, located in the cytoplasmic C-terminus of the H_1_ receptor, is essential [151]. Whether intracellular forces, such as NMMII-dependent contractions, influence the activities of heterotrimeric G proteins and GPCRs remains poorly understood.

### 4.4. Integrins

Integrins, heterodimeric adhesion proteins linking extracellular matrix proteins to the cytoskeleton, comprise 18 α- and 8 β-subunits, which form 24 heterodimers [152]. Integrins sense various mechanical stresses and convert a mechanical force into biochemical signaling within the cell [58]. The multiple roles of integrins—expressed on ECs, leukocytes, monocytes/macrophages, vascular smooth muscle cells, and platelets—in the pathogenesis of atherosclerosis are reviewed in depth elsewhere [153,154].

Integrins are sensitive to the lipid bilayer ordering and thickness [155,156,157,158]. Moreover, in PAECs and mouse embryonic fibroblasts, integrins themselves seem to increase the lipid bilayer order [155]. The exposure of HUVECs to pro-inflammatory oscillatory SS (0.5 ± 4 dyn/cm^2^) or anti-inflammatory pulsatile SS (12 ± 4 dyn/cm^2^) induced the opposite effects on α5β1 integrin localization in lipid rafts [156]. Oscillatory SS and pulsatile SS increase or decrease the levels of α5β1 integrins in lipid raft regions, respectively [156]. The exposure of HAECs to three non-lipid amphiphiles (vitamin E, Triton X-100, and benzyl alcohol) was used for the study of β1 integrin dependence on lipid bilayer order and domain thickness [157]. Only benzyl alcohol partitioned into the liquid-disordered domains and thinned these domains, enhancing β1 integrin affinity and valency, and inducing β1 integrin clustering [157]. Lietha and Izard suggested that mechanical stress-induced β integrin activation is mediated by membrane thinning [158].

Integrins are linked to the actin-based cytoskeleton via numerous scaffolding proteins, such as talin, vinculin, filamin A, and zyxin [58,152]. Integrins themselves regulate NMMII [153]. NMMII-generated contractility is required for the maintenance of integrin adhesion complexes [159]. In the stationary membranes of BAECs, the tension across vinculin was generated by NMMII and its upstream regulator RhoA, and was about 2.5 pN (0.25 μdyn) [62].

## 5. Conclusions

The experimental data considered here suggest that internal cellular forces—mainly NMMII contractility and actin-based lamellipodial protrusions, together with pro-inflammatory cholesterol- and oxidative stress-induced changes in the biomechanics of the lipid bilayer—tune the mechanosensitivity of the endothelial mechanosensors. Since RhoA and Rac1, key regulators of NMMII contractility and actin filaments assembly, are themselves critically regulated by pro-inflammatory agents, the biomechanics of the plasma membrane of ECs can be softened or stiffened upon inflammation, with subsequent alteration in ECs response to fluid SS. Fluid SS, depending on its patterns, is among the main controllers of endothelial inflammation. We propose that the altered—by first exposure of ECs to pro-inflammatory agents—mechanosensitivity of mechanosensors leads to their aberrant activation, and to the formation of a vicious cycle where pro-inflammatory signaling is sustained. Low-grade vascular inflammation then promotes atherogenesis.

## Figures and Tables

**Figure 1 membranes-12-00205-f001:**
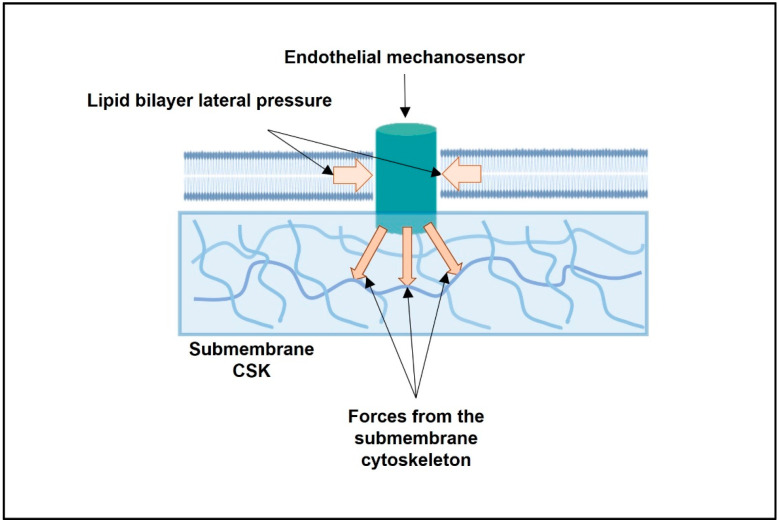
Scheme illustrating the actions of forces arising from the lipid bilayer and from the submembrane cytoskeleton and converging on a single mechanosensitive transmembrane protein.

**Figure 2 membranes-12-00205-f002:**
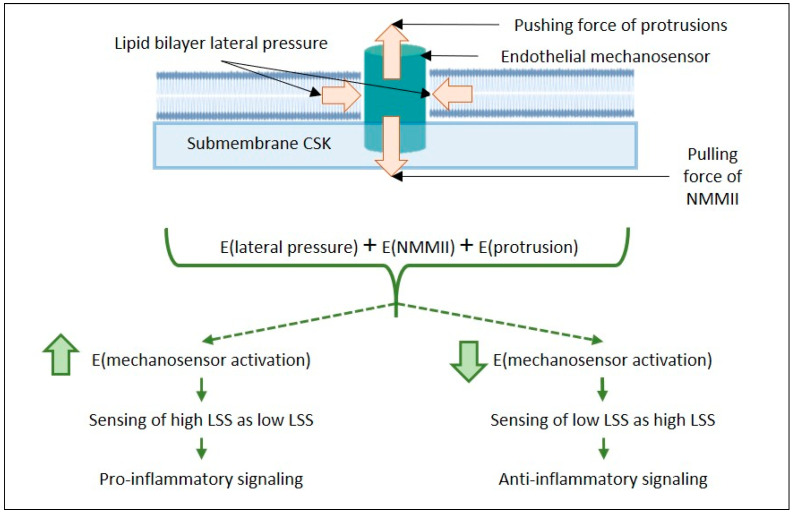
Scheme illustrating the actions of intracellular forces on any single endothelial mechanosensor. The transmembrane domain(s) of the mechanosensor is sensing the lipid bilayer lateral pressure. A pulling force from NMMII-generated tension and a pushing force from an actin-based protrusion are also sensed by the mechanosensor, meaning that the energy required for mechanosensor activation can be increased; in this case, the high LSS can be perceived as low LSS, or decreased, in which case the low LSS can be perceived as high LSS. Thus, the interplay of intracellular forces can shift LSS-induced signaling to be either pro-atherogenic or anti-atherogenic. Abbreviations: LSS, laminar shear stress; NMMII, non-muscle myosin II; smACSK, submembrane actin-based cytoskeleton.

**Figure 3 membranes-12-00205-f003:**
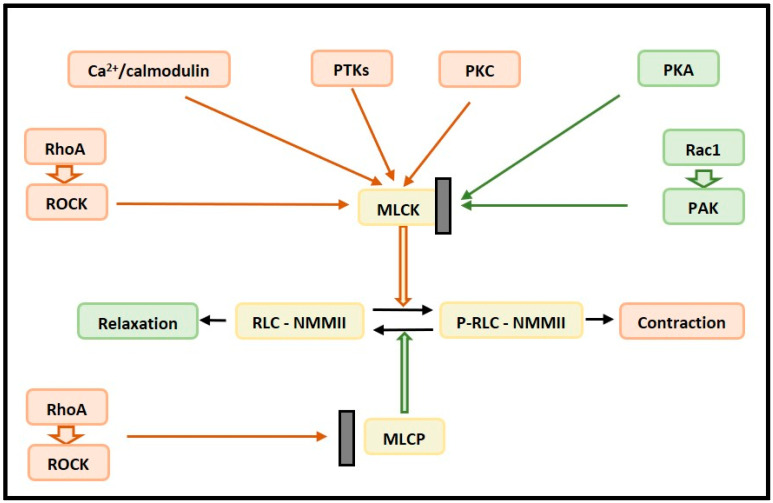
Scheme illustrating the regulation of NMMII contractility. MLCK and Rho–ROCK pathways lead to the phosphorylation of the RLCs of NMMII and NMMII contraction. MLCP and Rac1–PAK pathways lead to NMMII relaxation. MLCK itself is stimulated by upstream Ca^2+^/calmodulin, PTKs, and PKC, while PKA inhibits MLCK. Abbreviations: MLCK, myosin light chain kinase; MLCP, myosin light chain phosphatase; NMMII, non-muscle myosin II; PAK, p21-activated kinase; PKA, protein kinase A; PKC, protein kinase C; PTKs, protein tyrosine kinases; RLCs, regulatory light chains; ROCK, Rho-associated coiled-coil-containing kinase.

**Figure 4 membranes-12-00205-f004:**
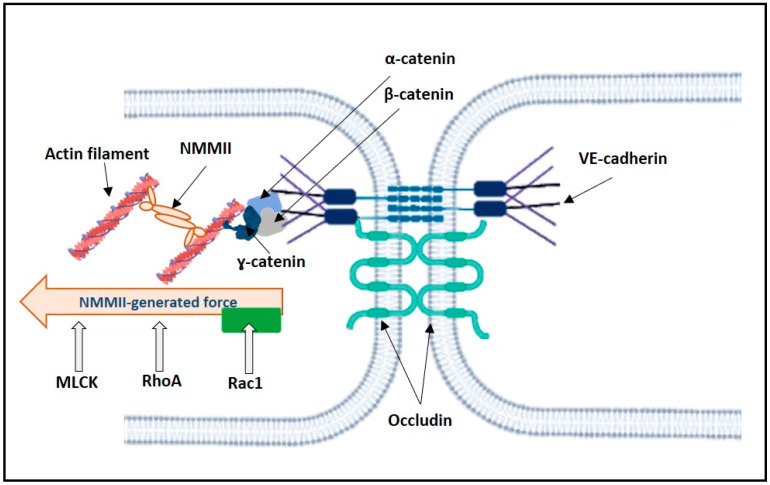
Scheme illustrating the regulation of NMMII-generated force acting on VE-cadherin. MLCK and RhoA lead to NMMII contraction. Rac1 leads to NMMII relaxation. Thus, the NMMII-generated force acting on VE-cadherin is controlled by counterbalance between MLCK, RhoA, and Rac1.

**Figure 5 membranes-12-00205-f005:**
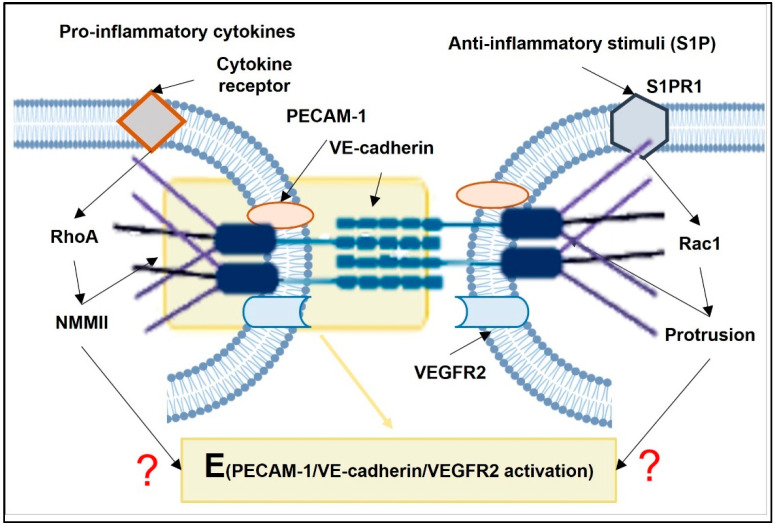
Scheme illustrating the actions of RhoA- and Rac1-dependent intracellular forces on the PECAM-1/VE-cadherin/VEGFR2 complex.

**Table 1 membranes-12-00205-t001:** Transmembrane and membrane-associated proteins under NMMII-generated tension.

Protein	Tension across Protein	NMMII Involvement	Cell Type	Reference
E-cadherin ^1^	1–2 pN	+	MDCK epithelial cells	[100]
PECAM-1 in static cells	negligible		BAECs	[63]
PECAM-1 in cells under SS	2.0 pN/molecule	Vimentin is involved	BAECs	[63]
VE-cadherin in static cells	2.4 nN/molecule	+	BAECs	[63]
VE-cadherin in cells under SS	1.8 nN/molecule	+	BAECs	[63]
Vinculin	~2.5	+	BAECs	[62]

^1^ Abbreviations: BAECs, bovine aortic endothelial cells; E-cadherin, epithelial cadherin; MDCK, Madin-Darby canine kidney; nN, nano-Newton; PECAM-1, platelet endothelial cell adhesion molecule 1; pN, pico-Newton; VE-cadherin, vascular endothelial cadherin.

**Table 2 membranes-12-00205-t002:** Hypoxia and oxidative stress in RhoA and Rac1 regulation in ECs. ↑—activation, ↓—inhibition.

Stress Factor	Effect on RhoA or Rac1	EC Type	Reference
Hypoxia	↑ RhoA	Porcine AECs ^1^	[93]
	↑ RhoA	Piglet PAEC	[116]
	↑ RhoA	Rat PAECs	[117]
	↓ Rac1	Porcine -AECs	[93]
	↓ Rac1	Piglet PAECs	[116]
Oxidative stress	↑ RhoA	Rat PAECs	[117]
	↑ Rac1	BAECs	[118]

^1^ Abbreviations: AECs, aortic endothelial cells; BAECs, bovine aortic endothelial cells; PAECs, pulmonary artery endothelial cells.

## Data Availability

Not applicable.

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
