# Peer review of "Endothelial Cell Plasma Membrane Biomechanics Mediates Effects of Pro-Inflammatory Factors on Endothelial Mechanosensors: Vicious Circle Formation in Atherogenic Inflammation"

_membranes, 2022, doi:10.3390/membranes12020205_

Round 1
Reviewer 1 Report
Review hypothesis
The authors present interesting hypothesis on differential impact of shear stress on the endothelial cells. In my opinion, the manuscript is worth to publish.
Nevertheless, I have few comments, that in my opinion could improve the article.
Major:
1. The authors should first provide the background information and in the separate chapter present their hypothesis. It will make clear division between the basis on which hypothesis is drawn and the results of the studies.
2. It would be useful if the thesis was presented on the scheme.
Minor
1. Illustration of information presented in the first part of the introduction, especially that starting in line 99, would be useful.
2. First part of the introduction have no title.
3. There is suffix 9, in the authors list. Ms Antonella Pantaleo. It is not explained.
4. Authors describe forces and pressure using two different units, dynes/sq cm or pN. Please unify.
Author Response
Review hypothesis
The authors present interesting hypothesis on differential impact of shear stress on the endothelial cells. In my opinion, the manuscript is worth to publish.
Nevertheless, I have few comments, that in my opinion could improve the article.
Major:
1. The authors should first provide the background information and in the separate chapter present their hypothesis. It will make clear division between the basis on which hypothesis is drawn and the results of the studies.
Authors: We would like to thank the reviewer #1 for his/her valuable comments on the manuscript. Our hypothesis is now in individual paragraphs (section 2: What intracellular forces are acting on any single transmembrane endothelial mechanosensor?) and therefore clearly separated from the basic review.
- It would be useful if the thesis was presented on the scheme.
Authors: To address the concerns raised, we have now added the graphical abstract illustrating the idea of the paper.
Minor
1. Illustration of information presented in the first part of the introduction, especially that starting in line 99, would be useful.
Authors: We appreciate the reviewer’s suggestion. We have added a new figure (Figure 1) illustrating the actions of forces arising from the lipid bilayer and from the submembrane cytoskeleton
- First part of the introduction have no title.
Authors: According to the reviewer’s suggestion, we have now subdivided the introduction into two subsections and titled them:
1.1. Blood flow patterns and pro-inflammatory response of endothelial cells
1.2. Forces and plasma membrane mechanosensors
- There is suffix 9, in the authors list. Ms Antonella Pantaleo. It is not explained.
Authors: We thank the reviewer for pointing this out. We now added the right suffix (6)
- Authors describe forces and pressure using two different units, dynes/sq cm or pN. Please unify.
Authors: We cannot unify pN and dynes/cm2, since they describe different phenomena, pN is used to measure the force acting on one single protein, while shear stress (dynes/cm2) deals with force acting tangentially to cell surface.
Authors: We thank the reviewer #1 for the constructive and insightful comments, which have helped us to substantially improve our manuscript
Reviewer 2 Report
Very heterogeneous and unfocused discussion of, in principle, interesting hypothesis with clear implications in pathophysiology. Too often the text is just a collection of facts without conclusions. The hypothesis is formulated in the title and in 1-2 sentences in the Conclusion; experimental works directly related to the hypothesis are not discussed or buried in the bulk of descriptive information. In other words, quality of presentation and discussion of the hypothesis is low.
Author Response
Very heterogeneous and unfocused discussion of, in principle, interesting hypothesis with clear implications in pathophysiology. Too often the text is just a collection of facts without conclusions. The hypothesis is formulated in the title and in 1-2 sentences in the Conclusion; experimental works directly related to the hypothesis are not discussed or buried in the bulk of descriptive information. In other words, quality of presentation and discussion of the hypothesis is low.
Authors: We would like to thank Reviewer #2 for taking the time and effort necessary to review our manuscript. As can be seen above and in the manuscript, we believe that the constructive criticism of reviewer 1 helped us to significantly improve our manuscript and we hope it now also satisfies reviewer 2.
Round 2
Reviewer 2 Report
Unfortunately, even after revision, the manuscript looks like a collection of facts on several topics without explaining links between them. The major hypothesis is formulated but absolutely not supported by discussion of experimental data directly addressing it. English has to be corrected.
Author Response
We Thank the reviewer for His/Her feedback, we believe it has improved our work on this manuscript.
As requested we have deeply revised the English lenguage and grammar throughout the paper in order to improve the flow and readability. In addition to this, our manuscript has been modified into a review paper rather than an hypothesis paper as suggested. We really hope that the applied modifications are in line with the expected changes requested in the latest revision.
Round 3
Reviewer 2 Report
I have no further comments.